# Development of a Decision Support Tool for Anticoagulation in Critically Ill Patients Admitted for SARS-CoV-2 Infection: The CALT Protocol

**DOI:** 10.3390/biomedicines11061504

**Published:** 2023-05-23

**Authors:** Victoria Dubar, Tiffany Pascreau, Annabelle Dupont, Sylvain Dubucquoi, Anne-Laure Dautigny, Benoit Ghozlan, Benjamin Zuber, François Mellot, Marc Vasse, Sophie Susen, Julien Poissy, Alexandre Gaudet

**Affiliations:** 1CHU Lille, Pôle de Médecine Intensive—Réanimation, F-59000 Lille, France; 2Biology Department, Hôpital Foch, F-92150 Suresnes, France; 3INSERM, Hémostase Inflammation Thrombose HITH U1176, Université Paris-Saclay, F-94276 Le Kremlin-Bicêtre, France; 4Hemostasis and Transfusion Department, Biology Pathology Center, University Hospital of Lille, F-59000 Lille, France; 5Institut d’Immunologie, Pôle de Biologie Pathologie Génétique Médicale, CHU Lille, F-59000 Lille, France; 6U1286-Institute for Translational Research in Inflammation (Infinite), Université de Lille, Inserm, CHU Lille, F-59000 Lille, France; 7Intensive Care Unit, Hôpital Foch, F-92150 Suresnes, France; 8Radiology Department, Hôpital Foch, F-92150 Suresnes, France; 9CNRS, Inserm U1285, UMR 8576—UGSF—Unité de Glycobiologie Structurale et Fonctionnelle, CHU Lille, Université de Lille, F-59000 Lille, France; 10CNRS, Inserm U1019-UMR9017-CIIL-Centre d’Infection et d’Immunité de Lille, Institut Pasteur de Lille, CHU Lille, Université de Lille, F-59000 Lille, France

**Keywords:** COVID-19, pulmonary embolism, immunothrombosis, prediction score, anticoagulation

## Abstract

Severe COVID-19 infections are at high risk of causing thromboembolic events (TEEs). However, the usual exams may be unavailable or unreliable in predicting the risk of TEEs at admission or during hospitalization. We performed a retrospective analysis of two centers (n = 124 patients) including severe COVID-19 patients to determine the specific risk factors of TEEs in SARS-CoV-2 infection at admission and during stays at the intensive care unit (ICU). We used stepwise regression to create two composite scores in order to predict TEEs in the first 48 h (H0–H48) and during the first 15 days (D1–D15) in ICU. We then evaluated the performance of our scores in our cohort. During the period H0–H48, patients with a TEE diagnosis had higher D-Dimers and ferritin values at day 1 (D1) and day 3 (D3) and a greater drop in fibrinogen between D1 and D3 compared with patients without TEEs. Over the period D1-D15, patients with a diagnosis of a TEE showed a more marked drop in fibrinogen and had higher D-Dimers and lactate dehydrogenase (LDH) values at D1 and D3. Based on ROC analysis, the COVID-related acute lung and deep vein thrombosis (CALT) 1 score, calculated at D1, had a diagnostic performance for TEEs at H0–H48, estimated using an area under the curve (AUC) of 0.85 (CI95%: 0.76–0.93, *p* < 10^−3^). The CALT 2 score, calculated at D3, predicted the occurrence of TEEs over the period D1-D15 with an estimated AUC of 0.85 (CI95%: 0.77–0.93, *p* < 10^−3^). These two scores were used as the basis for the development of the CALT protocol, a tool to assist in the decision to use anticoagulation during severe SARS-CoV-2 infections. The CALT scores showed good performances in predicting the risk of TEEs in severe COVID-19 patients at admission and during ICU stays. They could, therefore, be used as a decision support protocol on whether or not to initiate therapeutic anticoagulation.

## 1. Introduction

SARS-CoV-2 infections have been proven to be an important risk factor in the occurrence of thromboembolic events (TEEs), especially in patients admitted to intensive care units (ICU) [1]. It is, therefore, crucial to be able to identify TEE in severe COVID-19 cases, first of all, because of the frequency of TEEs in the severe forms of this disease and, secondly, because the current data do not allow us to confirm the use of systematic therapeutic anticoagulation in these patients. Thus, several clinical trials have shown that although a therapeutic anticoagulation protocol allows for a reduction in the occurrence of TEEs compared with a conventional strategy, it has no impact on mortality or the occurrence of organ failure. Moreover, the introduction of therapeutic anticoagulation is associated in these studies with a significant increase in the risk of bleeding events [2]. These data are in line with recommendations released in April 2020 and updated in 2021 [3] that propose using anticoagulation according to risk factors described in patients outside of COVID. This underlines the importance of individualizing the assessment of thrombotic risk from the perspective of a targeted initiation of therapeutic anticoagulation in severe COVID-19. 

Various studies have tried to understand the specificity of TEEs in COVID-19 patients, and it quickly appeared that coagulopathy resulting from SARS-CoV-2 infections is intimately linked with immunity activation in an immunothrombosis phenomenon [4]. This finding is paralleled by a change in the biological profile of patients as an increase in D-Dimers and fibrinogen [5].

The description of this phenomenon leads to questions about the accuracy of the usual thrombo-embolism markers, such as D-Dimers, which have often proved to be higher in SARS-CoV-2 infections [6]. Moreover, morphological exams are not always available for critical COVID-19 patients due to precarious respiratory status or acute kidney injury [7]. It is consequently a challenge to diagnose and predict the occurrence of TEEs both at admission and later during the ICU stay. Considering the pathophysiology of immunothrombosis, most studies have reported weak-to-moderate accuracies in the usual thromboembolic and immune markers for predicting the occurrence of TEEs in the setting of severe COVID-19 [8,9]. Some authors have proposed that endocan, a circulating proteoglycan secreted by pulmonary endothelial cells under pro-inflammatory conditions [10] whose high blood levels seem to correlate with pulmonary embolism (PE) [11], would be a useful biomarker of TEEs in COVID-19 [12].

In this retrospective study, we aimed to develop a COVID-related acute lung and deep vein thrombosis (CALT) protocol based on the usual clinical variables and on commonly available biomarkers for the prediction of TEEs in critically ill patients with severe COVID-19 infections. 

## 2. Materials and Methods

### 2.1. Study Population 

We retrospectively included patients hospitalized in the medical ICUs of two French hospitals (CHU de Lille and Hôpital Foch, Suresnes, France) between September 2020 and March 2022. Included patients met the following criteria: (1) age over 18; (2) admission to ICU for a COVID-19 infection confirmed with reverse transcriptase–polymerase chain reaction (RT-PCR); (3) the presence of respiratory support defining the severity of SARS-CoV-2 infection [3] within 48 h of admission to the ICU: high-flow nasal oxygen therapy (HFNO), continuous positive airway pressure (CPAP), non-invasive ventilation (NIV), invasive ventilation (IV), or extracorporeal membrane oxygenation (ECMO); (4) dosage of endocan; and (5) a thoracic-computed tomography (CT) scan with injection in the first 48 h of the ICU stay (H0–H48) in order to diagnose a potential PE. Patients with a positive RT-PCR at admission but with an alternative etiology for respiratory distress were not included in the study. 

### 2.2. Data Collection and Definitions

Patient demographic characteristics and comorbidities were recorded, as well as the occurrence of a TEE, defined by a diagnosis of PE confirmed by a thoracic CT scan or a diagnosis of deep vein thrombosis (DVT) confirmed using venous echo-Doppler. We defined the day of admission to the ICU (D1) as the first day of hospitalization in the ICU of one of the two centers of the study. Further data about the infection (vaccination status, PE or DVT at admission, variant) and about the therapeutics initiated in the first 48 h and during the ICU stay were also recorded. Blood samples were collected at baseline and on day 3 (D3). All biomarkers were assessed using automated standard methods.

Endocan assays were performed on EDTA plasma or on citrated plasma. A correction factor was applied for assays on a citrate tube to take into account the underestimation found with this type of matrix based on previously published pre-analytical data [13] according to the following rule: ×2 if the endocan us less than 3 ng/mL, ×1.5 if the endocan is between 3 and 5 ng/mL, ×1.25 if the endocan us between 5 and 8 ng/mL, and no correction if the endocan is >8 ng/mL.

We used the classification in the guidelines of the American Society of Hematology to define the intensity of anticoagulation (prophylactic, intermediate, or therapeutic) when collecting data from medical files [14].

### 2.3. Ethics Statement

The collection of the data analyzed in this study was approved by the Comité de Protection des Personnes Ile de France IV (approval number: 2020/30) for patients from the CHU de Lille and by the local ethics committee of the Foch Hospital for patients from Foch Hospital (approval number: 20-07-15). Written informed consent for the collection of data and biological samples was obtained from all the patients or their relatives. The management and analysis of data for the purposes of this study were declared to the French Commission Nationale de l’Informatique et des Libertés.

### 2.4. Statistical Analysis

#### 2.4.1. Exploratory Analysis

Categorical variables were expressed as numbers (percentages). Normally distributed continuous variables were expressed as means (SD). Non-normally distributed variables were expressed as median (IQR). Normality of distribution was checked graphically and by using the Shapiro–Wilk test. The association between the two variables was evaluated with a bivariate analysis. 

#### 2.4.2. Data Imputation

We performed a missing value imputation for continuous variables in the derivation cohort using the predictive mean-matching method with the *mice* function in R (*mice* R package) [15]. Counts of missing data for continuous variables are shown in Appendix A. Variables with missing data exceeding 35% of the total count were excluded from the analysis. 

#### 2.4.3. Derivation of CALT 1 and CALT 2

CALT 1 and CALT 2 scores were then constructed using multivariate logistic regression. We included the following explicative variables for the derivation of CALT 1: gender; age; weight; height; body mass index (BMI); admission simplified acute physiology score 2 (SAPS 2); admission sequential organ failure assessment (SOFA); diabetes; antiplatelet drug prior to hospitalization in the ICU; immunosuppression; vaccination status; variant; percentage and type of predominant COVID pulmonary lesions on the admission CT scan; treatments over the H0–H48 period using HFNO, CPAP, NIV, IV, ECMO; and the following biomarkers measured at D1: C-reactive protein (CRP), leukocytes, total lymphocytes, ferritin, procalcitonin (PCT), fibrinogen, D-Dimers, lactate dehydrogenase (LDH), and endocan.

The CALT 2 score was derived by including the same variables as for the CALT 1 score, as well as the following biomarkers measured at D3 and their respective variations from the values measured at D1: CRP, leukocytes, lymphocytes, fibrinogen, D-Dimers, ferritin, PCT, LDH, and endocan. For each score, we selected the predictor variables that contribute the most to the model using the stepwise regression method. We assessed the predictive performance of each model by calculating its area under receiver operating characteristics (AUROCs). The coefficients assigned to each selected variable were rounded to the nearest integer. We selected the predictor variables that contribute the most to the model using the smallest Akaike information criterion, allowing us to maximize the predictive performance while limiting model complexity. All statistical tests were two-tailed, and *p*-values < 0.05 were considered statistically significant. The statistical analyses were performed using R version 4.1.2 (R Foundation for Statistical Computing, Vienna, Austria).

## 3. Results

### 3.1. Patients’ Characteristics according to the Presence or Absence of TEEs 

Among the 1268 subjects with severe COVID-19 admitted to the ICUs at the participant centers, 124 met the inclusion criteria and were analyzed in our study. 

We compared the clinical characteristics and biomarkers between patients with and without a diagnosis of a TEE over the periods H0–H48 and D1–D15. The results of this analysis are shown in Table 1. 

Thirty-three (26%) patients were diagnosed with PE either at admission or during their stay in the ICU. Twenty-three patients (70%) were diagnosed during the H0–H48 period and ten (30%) were diagnosed between the third and fifteenth day (D3–D15) of hospitalization in the ICU. Two patients in the cohort developed DVT, both diagnosed within the first 48 h of their stays. Forty-four patients (35%) died during their stay in the ICU.

Patients with TEE at H0–H48 had a lower frequency of BMI > 30 kg/m^2^ than patients without TEE at H0–H48 (7 patients (32%) in the TEE group vs. 57 patients (61%) in the no-TEE group, *p* = 0.027), diabetes (4 patients (17%) in the TEE group vs. 42 patients (42%) in the no-TEE group, *p* = 0.038), and antiplatelet drugs prior to hospitalization in the ICU (no patients (0%) in the TEE group vs. 22 patients (27%) in the no-TEE group, *p* = 0.01). Moreover, therapeutic anticoagulation was more frequent in patients with TEEs at H0–H48 than in patients with no TEEs at H0–H48 (24 patients (100%) in the TEE group vs. no patients (0%) in the no-TEE group, *p* < 0.001). Furthermore, compared with patients without TEEs at D1–D15, patients with TEEs at D1-D15 had a lower frequency of BMI > 30 kg/m^2^ (12 patients (38%) in the TEE group vs. 52 patients (62%) in the no-TEE group, *p* = 0.031), diabetes (7 patients (21%) in the TEE group vs. 39 patients (43%) in the no-TEE group, *p* = 0.033), and antiplatelet drugs prior to hospitalization in the ICU (no patients (0%) in the TEE group vs. 22 patients (29%) in the no-TEE group, *p* = 0.001). Therapeutic anticoagulation was more frequent in patients with TEEs at D1-D15 than in patients with no TEEs at D1–D15 (26 patients (76%) in the TEE group vs. 15 patients (17%) in the no-TEE group, *p* < 0.001).

A comparison of biological variables according to the presence or absence of TEEs over the periods H0–H48 and D1–D15 is shown in Table 2. Compared with the group without TEEs at H0–H48, there was a significantly higher value in the group with TEEs at H0–H48 for D-Dimers at D1 (median (IQR): 2830 (1121; 4054) in the TEE group vs. 1100 (710; 2450) in the no-TEE group, *p* = 0.033), D-Dimers at D3 (median (IQR): 3590 (1549; 4000) in the TEE group vs 1214 (750; 1957) in the no-TEE group, *p* = 0.002), ferritin at D1 (median (IQR): 2304 (1373; 3497) in the TEE group vs. 1207 (684; 2149) in the no-TEE group, *p* = 0.006), ferritin at D3 (median (IQR): 2434 (1305; 3384) in the TEE group vs. 1241 (882; 2161) in the no-TEE group, *p* = 0.038), and LDH at D1 (median (IQR): 616 (490; 777) in the TEE group vs. (458 (357; 588) in the no-TEE group, *p*: 0.007). In addition, in patients with TEEs at H0–H48, we found lower fibrinogen at D3 (median (IQR): 5 (3.7; 5.9) in the TEE group vs. 5.8 (5.1; 7) in the no-TEE group, *p* = 0.006) and a greater decrease in fibrinogen between D1 and D3 (median (IQR): −1.7 (−2.3; −0.5) in the TEE group vs. −0.6 (−1.4, −0.1) in the no-TEE group, *p* = 0.006). 

Compared with the group without TEEs over the period D1-D15, there was a significantly higher value in the group with TEEs at D1-D15 for D-Dimers at D1 (median (IQR): 1991 (1048; 4000) in the TEE group vs. 1075 (715; 2293) in the no-TEE group, *p* = 0.025), D-Dimers at D3 (median (IQR): 3795 (1537; 4000) in the TEE group vs. 1176 (710; 1740) in the no-TEE group, *p* < 0.001), LDH at D1 (median (IQR): 616 (470; 769) in the TEE group vs. 458 (356; 587) in the no-TEE group, *p* = 0.004), LDH at D3 (median (IQR): 564 (454; 679) in the TEE group vs. 443 (349; 575) in the no-TEE group, *p* = 0.045), and ferritin at D1 (median (IQR): 1692 (1264; 3123) in the TEE group vs. 1175 (675; 2175) in the no-TEE group, *p* = 0.007). In addition, there was a greater decrease in fibrinogen between D1 and D3 in the TEE group (median (IQR: −1.6 (−2.2; −0.5) in the TEE group vs. −0.6 (−1.4; 0.1) in the no-TEE group, *p* = 0.02).

### 3.2. Construction of CALT 1 and CALT 2 Scores

We then used bidirectional stepwise logistic regression to construct predictive scores for the diagnosis of TEEs in our study population. This method allowed us to develop (1) the CALT 1 score, calculated at D1, aimed at predicting the existence of a TEE diagnosed during the period H0–H48, and designed as a decision support tool for anticoagulation at D1; (2) the CALT 2 score, calculated at D3, aimed at predicting the existence of a TEE diagnosed during the period D1-D15, and designed as a decision support tool for anticoagulation at D3. 

#### 3.2.1. Construction and Evaluation of the CALT 1 Score

The variables and coefficients used in the construction of CALT 1 are presented in Figure 1A. 

The performance of the CALT 1 score was evaluated through its ROC curve, which was compared with those of all the parameters constituting this score. The AUROC of the CALT 1 score to predict the diagnosis of TEEs in the H0–H48 period was calculated at 0.85 (CI95%: 0.76–0.93, *p* < 10^−3^). In comparison, the best-performing single variable was ferritin, whose AUROC was calculated at 0.69 (CI95%: 0.58–0.8, *p* = 0.009) (Figure 1B). In addition, the values taken by CALT 1 according to the presence or absence of TEEs over the H0–H48 period are shown in Figure 1C, and the intrinsic performances of CALT 1 (sensitivity and specificity), as well as the positive (PPV) and negative (NPV) predictive values in our cohort, are described in Appendix A. Based on these results, we identified two thresholds of interest for CALT 1 that could be considered in an anticoagulation decision support algorithm. Indeed, we observed that a CALT 1 value strictly lower than 2 was associated with a specificity of 69%, meaning that it correctly identified 69 out of 100 patients without TEEs at H0–H48 in our cohort, with an NPV at 95%. Furthermore, a CALT 1 value greater than or equal to 4 had a sensitivity of 50%, meaning that it was able to detect 12 out of 24 patients with TEE at H0–H48 in our cohort, with a PPV of 74%.

#### 3.2.2. Construction and Evaluation of the CALT 2 Score

Concerning the CALT 2 score, the variables and coefficients retained for its construction are shown in Figure 2A. As for CALT 1, we evaluated the accuracy of CALT 2 by calculating its AUROC, which we compared with those of its constitutive parameters. The AUROC of CALT 2 to predict the diagnosis of TEEs over the period D1-D15 was calculated at 0.85 (CI95%: 0.77–0.93, *p* < 10^−3^). D-Dimers measured at D3 were identified as the single parameter with the best performance, with an AUROC calculated at 0.72 (CI95%: 0.62–0.82, *p* < 10^−3^) (Figure 2B). The values taken by CALT 2 according to the presence or absence of TEEs over the period D1-D15 are shown in Figure 2C. The sensitivity, specificity, PPV, and NPV of CALT 2 in our cohort are shown in Appendix A.

Similar to the first part of the protocol, we retained two thresholds of clinical interest for the CALT 2 score. A CALT 2 value strictly lower than 0 had a specificity of 58%, meaning it allowed for the detection of 52 out of 90 patients with no TEE diagnosis at D1-D15, with an NPV at 97%. Additionally, a CALT 2 value greater than or equal to 3 yielded a sensitivity of 59%, meaning it detected 20 out of 34 patients with a TEE diagnosis at D1-D15, with a PPV of 72%.

#### 3.2.3. Proposal of a Decision Support Algorithm for Anticoagulation

Based on these results, we propose a decision algorithm, the CALT protocol, described in Figure 3. Following this protocol, the calculation of the CALT 1 score is performed at D1 and helps in deciding about the initiation of anticoagulation if the score is greater than or equal to four. If the score is strictly less than two, the protocol suggests not initiating therapeutic anticoagulation. In the “gray area” between these two thresholds, the protocol does not make any specific suggestion regarding anticoagulation. The protocol then relies on the calculation of the CALT 2 score at D3 to assess the probability of a TEE diagnosis over the period D1-D15. In cases of a CALT 2 score greater than or equal to three, associated with a high risk of TEE diagnosis over the period D1-D15, the protocol suggests that therapeutic anticoagulation should be maintained or introduced on D3. In the case of a CALT 2 score strictly below zero, the protocol suggests that therapeutic anticoagulation should not be maintained or introduced. A CALT 2 score between these two limits falls into the “gray zone”, for which no recommendation regarding anticoagulation is provided.

## 4. Discussion

The identification of the predictive markers of TEEs appears to be relevant for severe SARS-CoV-2 pneumonia as the use of a CT scan may be compromised by the severity of the patient, possible contra-indications, or the unavailability of the technique. Thus, in a previous study, 30% of patients did not have a CT scan during hospitalization in the ICU [7]. The main purpose of this study was to propose a practical tool to help the clinician in the decision to start or maintain therapeutic anticoagulation in severe COVID-19. 

Few studies, to the best of our knowledge, have investigated markers that are predictive of TEEs in patients admitted to the ICU with a SARS-CoV-2 infection. Furthermore, most studies have focused on markers previously evaluated in venous thromboembolic disease. Thus, the use of D-Dimers has been assessed in several studies, both as a marker of TEEs [16] and mortality [6]. However, their lack of specificity makes them unreliable as a standalone diagnostic marker, particularly in the setting of intensive care [17]. Our study identified several factors associated with the risk of TEEs in the first 15 days after ICU admission. Concerning the risk of TEEs in the H0–H48 period, obesity, diabetes, and prior antiplatelet treatment were less frequently found in the group with TEEs. Higher D-Dimer values and an inflammatory biological profile at admission (high ferritin and LDH values) were associated with the diagnosis of TEEs at H48. 

The identification of diabetes, obesity, and antiplatelet treatments as protective factors was surprising. Indeed, previously published results suggested a positive association between TEEs and obesity [18] or diabetes [19]. However, the data are divergent, and other studies did not find a significant association [20]. Furthermore, the absence of a protective effect from antiplatelet treatments in preventing TEEs in the setting of severe COVID-19 was reported in a recent work [21]. Concerning the association between TEEs and biological parameters, our study confirmed the predictive character of D-Dimer levels, as demonstrated in previous studies. We also found an association between increased inflammatory parameters (ferritin and LDH) and TEEs. This correlation between high inflammatory parameters and the occurrence of TEEs was consistent with other results reporting an association between the incidence of TEEs and either higher ferritin [22] or CRP and neutrophil levels [23]. Interestingly, we found a more pronounced decrease in fibrinogen levels in subjects who developed TEEs, as previously reported in the literature [24,25]. This result may be a reflection of activated coagulation in patients with TEEs, leading to fibrinogen consumption. However, the interference of potential confounders such as ECMO, which is known to lower fibrinogen blood levels, cannot be ruled out. 

In order to make the connection between these epidemiological observations and our clinical practice, we developed a tool to help in the decision to initiate anticoagulation, relying on two scores, CALT 1 and 2, allowing us to estimate thromboembolic risk during stays in the ICU. 

The models derived from our logistic regression analyses for the CALT 1 and CALT 2 scores consistently show an association between elevated inflammatory parameters and the diagnosis of TEEs. The leukocyte count and ferritin are thus found in these scores to be predictors of the diagnosis of TEEs. Regarding CALT 1, it may seem surprising that few variables were finally retained in the final model obtained using stepwise regression. However, it must be noted that this method looks for variables that contribute the most to the model when combined. Therefore, in cases of high collinearity between multiple factors, only the most useful for the model is finally retained. Accordingly, in our cohort, fibrinogen, D-Dimer, and LDH levels provided similar information to ferritin, explaining why, among all these variables with significant values between groups, only ferritin was finally retained to increase the accuracy of the CALT 1 score. Similarly, LDH was excluded from the final model for CALT 2 because of redundancy with other parameters included in the score, even though it may seem more clinically relevant.

More unexpectedly, CALT 1 and 2 incorporate elevated endocan, previously described as a marker of lung inflammation, as a protective factor. This marker was previously studied in severe COVID as a prognostic marker [7], but to the best of our knowledge, it has never been evaluated as a thromboembolic marker in this context. It should be noted that the use of endocan for the diagnosis of PE outside COVID was suggested in a previous study [17], leading some authors to hypothesize that this association could be found in the context of SARS-CoV-2 infections. However, our study seems to refute this hypothesis. Procalcitonin appears, in our study, to be negatively associated with the risk of TEEs. This result can be explained by the predictive value of procalcitonin for bacterial infections. This association was suggested by the results of a large retrospective study on COVID-19 patients, which found higher procalcitonin values in cases of bacterial co-infection, even though the diagnostic accuracy of procalcitonin in this setting remained very low [26]. However, in cases of COVID requiring transfer to the ICU, a high procalcitonin value may increase the probability of bacterial superinfection, thus decreasing the probability of a thromboembolic hypothesis as a cause of the aggravation. Age is also a predictor of TEEs at admission in our model, which appears consistent with its status as an important risk factor for venous thromboembolic disease both in the general population and in patients infected with SARS-CoV-2 [27]. Several potential benefits of the CALT protocol can be underlined for clinical practice. Firstly, the inclusion of additional explanatory variables appears to provide greater accuracy than the use of each factor individually, as illustrated by the ROC curves presented in our results. In addition, the use of CALT 1 and CALT 2 scores could be useful in guiding the level of anticoagulation in patients admitted to the ICU for SARS-CoV-2 infections via a two-step strategy. First, CALT 1, calculated at admission to the ICU, could be considered in order to decide whether or not to initiate therapeutic anticoagulation for the first 48 h for patients in whom a chest CT scan cannot be performed at ICU admission. This situation seems to concern a non-negligible proportion of patients admitted to the ICU for COVID, as previously described in our center [7]. Alternatives to CT scans have been studied, such as endobronchial ultrasound (EBUS) in patients with ECMO [28] or ventilation/perfusion (V/P) scintigraphy [27]. However, these techniques remain difficult to access for critically ill patients and are not widely used in current practice. In the second phase, on the third day of hospitalization, the calculation of CALT 2 seems to allow for the identification of patients who should benefit from therapeutic anticoagulation because of a hypercoagulable state, at high risk of constituted or future thrombosis. Notably, the construction of CALT 2 includes the scope of CALT 1, i.e., the prediction of TEEs over the period H0–H48, as it looks for TEEs over the period D1-D15 rather than D3-D15. This reassessment at D3 of the overall probability of TEEs over the period D1-D15 seems to be of particular interest, as it allows the anticoagulation to be adjusted to the evolution of the thromboembolic risk. In detail, we found in six patients a CALT 1 lower than two at D1, indicating a low thromboembolic risk at D1, progressing toward a high thromboembolic risk at the re-evaluation at D3, reflected by a CALT 2 greater than or equal to three. Conversely, however, only one patient showed a transition from high thromboembolic risk at admission (CALT 1 greater than or equal to four) to low at the reassessment at D3 (CALT 2 less than zero).

Our study has several strengths. It is a bicentric study covering a large period over several epidemic waves during which practices evolved. Most of the parameters evaluated are used in common practice. We extended our investigations to factors that are less widely used but of potential interest in view of the pathophysiology of the coagulopathy observed during severe SARS-CoV-2 infections. We also propose a practical algorithm for routine clinical practice based on a two-stage decision score that can be used on patients on whom CT scans cannot be performed. 

One of the main limitations of our study is the absence of data before the current hospitalization. We included a heterogeneous population, including subjects coming from home, from conventional hospitalization, or from another ICU, most often in order to benefit from ECMO. As such, we could not collect reliable data on the existence of previous thromboprophylaxis or therapeutic anticoagulation. This limitation can be analyzed in light of the data reported by Battistoni et al., who showed that the use of chronic anticoagulation was not associated with an overall increase in ICU mortality or morbidity [29]. However, in the adjusted survival analysis of this study, in-hospital anticoagulation was associated with higher survival compared with no anticoagulation, supporting the hypothesis that anticoagulation started during hospitalization has a benefit. Notably, we were only able, in our study, to collect data regarding the prophylactic or therapeutic nature of anticoagulation for the first 48 h in the ICU and found, as expected, that all patients with PE upon admission to the ICU initially received therapeutic anticoagulation. However, these data do not allow us to prejudge with certainty the nature of anticoagulation in our cohort beyond the initial phase of ICU management. This limitation can be tempered for several reasons. First, anticoagulation is frequently stopped in the ICU, whether for invasive procedures, transport, or because of the occurrence of bleeding events. Secondly, a study by Novelli et al. highlights the frequent discrepancy between the target and the level of anti-Xa activity actually achieved during therapeutic anticoagulation [30].

A second limitation concerns the retrospective nature of the study, which resulted in missing data (see Appendix A). Nevertheless, our score constructions were restricted to variables for which the missing data represented only a limited fraction of the total number of patients in our cohort. Moreover, as these were retrospective data, a CT scan looking for a TEE diagnosis was not systematically performed for all patients at D15, which may have underestimated the proportion of TEEs during the stay in patients who progressed favorably. This may have led to false negatives in CALT 2. Regarding CALT 1, however, it can be emphasized that a CT scan result within the first 48 h was an inclusion criterion of our study, resulting in good reliability for the incidence of TEEs at admission. 

The third limitation lies in the existence of a gray zone for the CALT scores, with 35/124 subjects (28%) at D1 and 43/124 subjects (35%) at D3 for whom our protocol was unable to predict the risk of TEEs. This proportion remains, nevertheless, limited, allowing a majority of patients to benefit from a therapy that appears to be adapted to their level of risk.

The fourth limitation results from the fact that blood-based endocan measurement is generally not available as part of routine clinical practice in most centers. Thus, the use of risk stratification and decision-making provided by CALT 1 and CALT 2 score results, based on endocan measurements, may appear to be of limited impact. This should be analyzed in light of the somewhat moderate contribution of the variable endocan in the performances of the CALT protocol, with AUROCs for CALT 1 and CALT 2, respectively, falling at 0.8 and 0.78 when omitting endocan for the calculation of these scores.

Another important limitation results from the striking changes in the epidemiology of SARS-CoV-2 infections since the appearance of the outbreak [31]. Treatments and vaccination have deeply modified the epidemiology of COVID-19 in the ICU by reducing the incidence of severe SARS-CoV2 infections. The proportion of immunocompromised patients in severe cases has strongly increased, questioning the validity of our protocol in this new setting. Further studies will be necessary to assess the performance of the protocol in the immunocompromised population. 

One last limitation of our study is the absence of a validation cohort to verify whether our results can be reproduced. Additional assessments of our protocol in independent cohorts of patients are warranted to support our conclusions.

## 5. Conclusions

In patients with severe SARS-CoV-2 infections, we identified specific risk factors of TEEs at admission and during ICU stays. Based on these results, we were able to establish a tool to assist the decision-making related to therapeutic anticoagulation during severe forms of SARS-CoV-2 infection: the CALT protocol. These results must be considered from the perspective of the changing epidemiological features of severe COVID-19, and therefore, further studies are required to test the validity of our protocol.

## Figures and Tables

**Figure 1 biomedicines-11-01504-f001:**
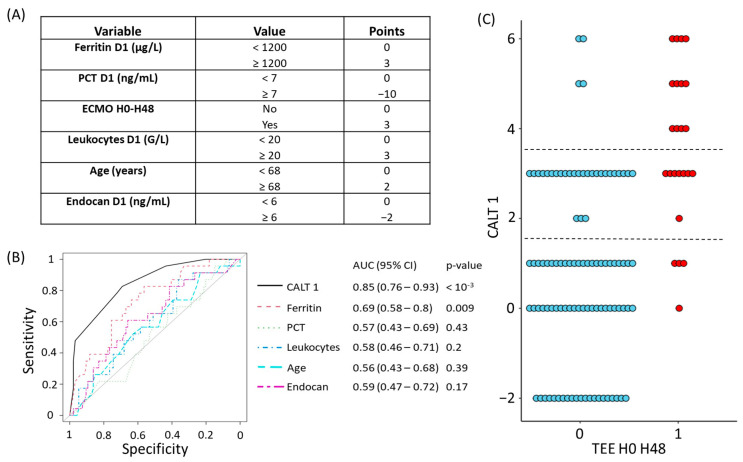
Description and evaluation of the accuracy of the CALT 1 score for the diagnosis of TEEs at H0–H48. (**A**) Variables retained for the construction of the CALT 1 score. (**B**) ROC curves of the CALT 1 score and its constitutive variables for the diagnosis of TEEs over the H0–H48 period. (**C**) Values of the CALT 1 score according to the presence or absence of a TEE at H0–H48. Red dots represent patients with TEEs diagnosed in the H0–H48 period. Blue dots represent patients without a TEE diagnosis in the H0–H48 period. The dotted lines represent the low and high decision thresholds used for the CALT 1 score. AUC: area under the curve, CALT: COVID-related acute lung and deep venous thrombosis, ECMO: extracorporeal membrane oxygenation, TEE: thrombo-embolic event, PCT: procalcitonin.

**Figure 2 biomedicines-11-01504-f002:**
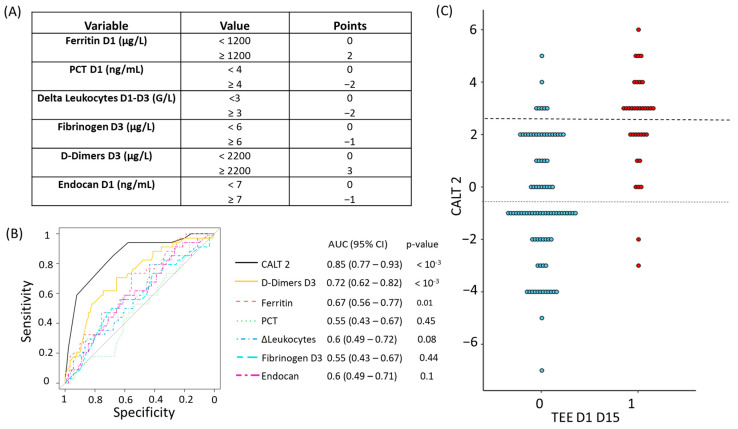
Description and evaluation of the accuracy of the CALT 2 score for the prediction of TEEs at D1-D15. (**A**) Variables retained for the construction of the CALT 2 score. (**B**) ROC curves of the CALT 2 score and its constitutive variables for the prediction of TEEs over the D1-D15 period. (**C**) Values of the CALT 2 score according to the presence or absence of TEEs at D1-D15. Red dots represent patients with TEEs diagnosed in the D1-D15 period. Blue dots represent patients without a TEE diagnosis in the D1-D15 period. The dotted lines represent the low and high decision thresholds used for the CALT 2 score. AUC: area under the curve, CALT: COVID-related acute lung and deep venous thrombosis, ECMO: extracorporeal membrane oxygenation, TEE: thrombo-embolic event, PCT: procalcitonin.

**Figure 3 biomedicines-11-01504-f003:**
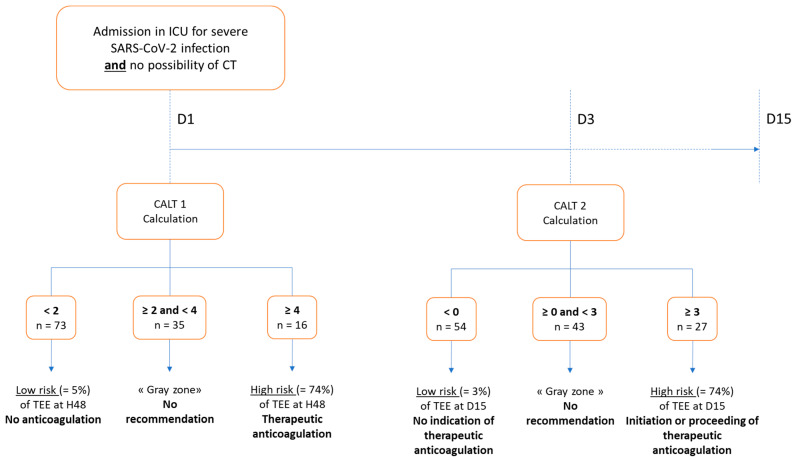
Description of the CALT protocol. Therapeutic anticoagulation is defined in accordance with the classification released by the American Society of Hematology, e.g., unfractionated heparin targeting anti-Xa activity at 0.3–0.7 IU/mL [14]. CALT: COVID-related acute lung and deep venous thrombosis, CT: computed tomography, TEE: thrombo-embolic event.

**Table 1 biomedicines-11-01504-t001:** Clinical characteristics associated with TEEs.

	TEE H0–H48		*p*	TEE D1–D15		*p*
	No (n = 100)	Yes (n = 24)		No (n = 90)	Yes (n = 34)	
**Demographics**						
Gender (male), n (%), *n′ = 124*	68 (68)	17 (71)	0.98	60 (67)	25 (74)	0.60
Age (years), mean (SD), *n′ = 124*	61 (11)	62 (11)	0.49	61 (11)	61 (11)	1
Comorbidities,						
BMI (kg/m^2^), mean (SD), *n′* = 116	31 (7)	31 (7)	0.61	32 (7)	30 (6)	0.22
BMI > 30 kg/m^2^, n (%), *n′* = 116	57 (61)	7 (32)	0.027	52 (62)	12 (38)	0.031
Diabetes, n (%), *n′* = 124	42 (42)	4 (17)	0.038	39 (43)	7 (21)	0.033
Chronic respiratory failure, n (%), *n′* = 124	7 (7)	0 (0)	0.4	7 (8)	0 (0)	0.22
COPD, n (%), *n′* = 124	4 (4)	1 (4)	1	4 (4)	1 (3)	1
Chronic heart failure, n (%), *n′* = 124	7 (7)	0 (0)	0.4	7 (8)	0 (0)	0.22
Cirrhosis (Child B or C), n (%), *n′* = 124	1 (1)	0 (0)	1	1 (1)	0 (0)	1
End-stage kidney disease, n (%), *n′* = 124	4 (4)	0 (0)	0.72	4 (4)	1 (3)	0.5
Immunocompromised, n (%), *n′* = 124	14 (14)	2 (8)	0.69	14 (16)	2 (6)	0.26
Antiplatelet drug prior to hospitalization, n (%), *n′ = 100*	22 (27)	0 (0)	0.01	22 (29)	0 (0)	0.001
SAPS 2, mean (SD), *n′ = 123*	41 (14)	43 (17)	0.45	41 (14)	42 (15)	0.58
SOFA, mean (SD), *n′ = 123*	5 (4)	6 (4)	0.36	5 (4)	6 (4)	0.28
**SARS-CoV-2 infection**						
Variant, *n′* = 103			0.36			0.83
Wild, n (%)	53 (63)	13 (68)		48 (65)	18 (62)	
Alpha, n (%)	16 (19)	5 (26)		14 (19)	7 (24)	
Delta, n (%)	15 (18)	1 (5)		12 (16)	4 (14)	
Vaccination status, *n′* = 100			0.56			0.67
Not vaccinated, n (%)	87 (91)	21 (88)		77 (90)	31 (91)	
Complete scheme, n (%)	2 (2)	0 (0)		2 (2)	0 (0)	
Incomplete scheme, n (%)	7 (7)	3 (13)		7 (8)	3 (9)	
Symptoms onset—ICU admission (days), mean (SD), *n′ = 121*	9 (6)	9 (5)	0.97	9 (5)	9 (5)	0.76
Extension of lung injury on CT scan (%), mean (SD), *n′ = 105*	52 (21)	57 (20)	0.41	51 (21)	58 (10)	0.16
Predominant findings in CT, *n′ = 98*			1			0.71
Consolidation, n (%)	58 (73)	14 (74)		55 (71)	20 (77)	
Ground-glass opacities, n (%)	42 (27)	10 (26)		35 (29)	14 (23)	
**Treatments during ICU hospitalization**						
Corticosteroids, n (%), *n′ = 124*	83 (83)	19 (80)	0.89	77 (86)	25 (74)	0.19
Remdesivir, n (%), *n′ = 124*	9 (9)	1 (4)	0.72	2 (2)	1 (3)	1
Tocilizumab, (n%), *n′ = 123*	2 (2)	1 (4)	1	7 (8)	3 (9)	1
HFNO H0–H48, n (%), *n′ = 124*	57 (57)	14 (58)	1	52 (58)	19 (56)	1
CPAP H0–H48, n (%), *n′ = 124*	28 (28)	7 (29)	1	27 (30)	8 (24)	0.62
NIV H0–H48, n (%), *n′ = 124*	25 (25)	4 (17)	0.55	22 (24)	7 (21)	0.83
IV H0–H48, n (%), *n′ = 124*	50 (50)	12 (50)	1	45 (50)	17 (50)	1
ECMO H0–H48, n (%), *n′ = 124*	12 (12)	6 (25)	0.19	10 (11)	8 (24)	0.14
Prone positioning H0–H48, n (%), *n′ = 100*	19 (19)	4 (17)	1	16 (18)	7 (21)	0.92
Antibiotherapy H0–H48, n (%), *n′ = 124*	66 (66)	17 (71)	0.83	59 (65)	24 (71)	0.75
Anticoagulation H0–H48, *n′ = 121*			<0.001			<0.001
Prophylactic dose, n (%)	60 (62)	0 (0)		56 (64)	4 (12)	
Intermediate dose, n (%)	20 (21)	0 (0)		16 (18)	4 (12)	
Therapeutic dose, n (%)	17 (18)	24 (100)		15 (17)	26 (76)	
**Outcomes**						
Duration of IV (days), mean (SD), *n′ = 120*	14 (15)	17 (20)	0.43	13 (15)	19 (18)	0.063
Duration of antibiotherapy (days), mean (SD), *n′ = 123*	11 (11)	12 (13)	0.76	10 (11)	13 (12)	0.27
ICU length of stay (days), mean (SD), *n′ = 123*	19 (18)	21 (22)	0.58	18 (19)	22 (20)	0.3
Mortality at D28, n (%), *n′ = 124*	34 (34)	4 (17)	0.16	31 (34)	7 (21)	0.2
Mortality at ICU discharge, n (%, *n′ = 124*)	38 (38)	6 (25)	0.34	33 (37)	11 (32)	0.81

Results for quantitative variables are expressed as mean with standard deviation (SD). Qualitative variables are expressed by their number and percentage of the total number of patients excluding missing data. *n′*: counts of non-missing data. BMI: body mass index, COPD: chronic obstructive pulmonary disease, CPAP: continuous positive airway pressure, ECMO: extracorporeal membrane oxygenation, HFNO: high-flow nasal oxygen, IV: invasive ventilation, NIV: non-invasive ventilation, SAPS 2: Simplified Acute Physiology Score 2, SOFA: sequential organ failure assessment, TEE: thrombo-embolic event.

**Table 2 biomedicines-11-01504-t002:** Biomarkers associated with TEEs.

	TEE H0–H48		*p*	TEE D1–D15		*p*
	No (n = 100)	Yes (n = 24)		No (n = 90)	Yes (n = 34)	
CRP (mg/L), median (IQR)						
D1, *n′ = 123*	133 [68; 186]	156 [81; 211]	0.30	130 [70; 196]	150 [83; 182]	0.48
D3, *n′ = 101*	67 [37;135]	45 [20; 124]	0.47	66 [34; 120]	58 [36; 153]	0.94
Delta, *n′ = 101*	−50 [−125; 21]	−53 [−128; 39]	0.86	−57 [−125; 8]	−48 [−126; 45]	0.52
Leukocytes (G/L), median (IQR)						
D1, *n′ = 124*	9 [5.7; 12]	10.2 [7;14]	0.2	9 [6; 12]	9 [6; 12]	0.22
D3, *n′ = 107*	10 [8; 13]	13 [9; 15]	0.14	11 [8; 13]	11 [8; 13]	0.49
Delta, *n′ = 107*	1 [−1; 5]	2 [−3; 3]	0.71	1 [−1; 5]	1 [−1; 5]	0.27
Lymphocytes (G/L), median (IQR)						
D1, *n′ = 98*	0.6 [0.4; 0.9]	0.6 [0.5; 0.9]	0.66	0.6 [0.4; 0.9]	0.6 [0.5;0.9]	0.49
D3, *n′ = 64*	0.8 [0.5; 1.1]	0.7 [0.4; 1.2]	0.64	0.8 [0.5; 1.1]	0.6 [0.4; 0.9]	0.071
Delta, *n′ = 64*	0 [−0.2; 0.5]	0.2 [0; 0.3]	0.76	0.2 [−0.1; 0.5]	−0.1 [−0.3; 0.2]	0.058
Platelets (G/L), median (IQR)						
D1, *n′ = 97*	237 [193; 308]	254 [201; 319]	0.4	239 [193; 306]	251 [193; 341]	0.45
D3, *n′ = 92*	276 [222; 344]	303 [206; 400]	0.51	278 [218; 350]	270 [213; 378]	0.84
Delta, *n′ = 92*	60 [−9; 103]	31 [−40; 69]	0.26	60 [−9; 105]	22 [−51; 66]	0.07
Fibrinogen (g/L), median (IQR)						
D1, *n′ = 116*	6.8 [5.9; 7.6]	6.3 [5.3; 7.4]	0.32	6.8 [5.8; 7.6]	6.4 [5.7; 7.4]	0.7
D3, *n′ = 87*	5.8 [5.1; 7]	5 [3.7; 5.9]	0.006	5.7 [5; 7]	5.4 [3.9; 6.2]	0.11
Delta, *n′ = 87*	−0.6 [−1.4. −0.1]	−1.7 [−2.3; −0.5]	0.006	−0.6 [−1.4; 0.1]	−1.6 [−2.2; −0.5]	0.02
D-Dimers (µg/L), median (IQR)						
D1, *n′ = 103*	1100 [710; 2450]	2830 [1121; 4054]	0.033	1075 [715; 2293]	1991 [1048; 4000]	0.025
D3, *n′ = 81*	1215 [710;1957]	3590 [1549; 4000]	0.002	1176 [710; 1740]	3795 [1537; 4000]	<0.001
Delta, *n′ = 81*	0 [−575; 392]	−75 [−3089; 1348]	0.77	0 [−566; 360]	−75 [−3089; 1348]	0.81
Ferritin (µg/L), median (IQR)						
D1, *n′ = 100*	1207 [684; 2149]	2304 [1373; 3497]	0.006	1175 [675; 2175]	1692 [1264; 3123]	0.007
D3, *n′ = 74*	1241 [882; 2161]	2434 [1305; 3384]	0.038	1223 [853; 2096]	1468 [1115; 3210]	0.062
Delta, *n′ = 74*	−197 [−787; −24]	284 [−557; 543]	0.12	−227 [−801; −35]	122 [−430; 543]	0.057
LDH (IU/L), median (IQR)						
D1, *n′ = 76*	458 [357; 588]	616 [490; 777]	0.007	458 [356; 587]	616 [470; 769]	0.004
D3, *n′ = 59*	443 [358; 591]	564 [465; 676]	0.084	443 [349; 575]	564 [454; 679]	0.045
Delta, *n′ = 59*	−10 [−64; 66]	−40 [−155; 23]	0.6	−11 [−63; 59]	−21 [−113; 34]	0.67
PCT (ng/mL), median (IQR)						
D1, *n′ = 100*	0.3 [0.1; 1.2]	0.2 [0.1; 0.8]	0.71	0.3 [0.1; 1.2]	0.2 [0.2; 0.7]	0.73
D3, *n′ = 79*	0.2 [0.1; 0.6]	0.2 [0.1; 0.3]	0.48	0.2 [0.1; 0.6]	0.2 [0.1; 0.4]	0.87
Delta, *n′ = 79*	−0.1 [−0.5; −0]	−0.1 [−0.3; 0]	0.74	−0.1 [−0.5; 0]	−0.2 [−0.5; 0]	0.69
Endocan (ng/mL), median (IQR)						
D1, *n′ = 124*	6.7 [4; 13]	4.8 [3.6; 10.2]	0.17	6.7 [4; 14.6]	6 [3. 11]	0.1
D3, *n′ = 60*	7.3 [4; 14]	9.2 [4; 21]	0.28	7 [4.8; 14]	11 [5.4.;20]	0.19
Delta, *n′ = 60*	0.4 [−2; 3]	0.3 [−1.3; 11]	0.59	0 [−3.2; 2.8]	1.8 [−0.2;7.8]	0.11

The variables are expressed as the median and interquartile range (IQR). The difference in value between D1 and D3 is expressed in absolute value. *n*′: counts of non-missing data. CRP: C-reactive protein, LDH: lactate dehydrogenase, PCT: procalcitonin, TEE: thrombo-embolic event.

## Data Availability

The datasets used and/or analyzed during the current study are available from the corresponding author upon reasonable request.

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
