# Peer review of "Development of a Decision Support Tool for Anticoagulation in Critically Ill Patients Admitted for SARS-CoV-2 Infection: The CALT Protocol"

_biomedicines, 2023, doi:10.3390/biomedicines11061504_

Round 1
Reviewer 1 Report
In a retrospective study, the authors aimed to develop the CALT protocol, based on usual clinical variables and on routinely available biomarkers for the prediction of thrombo-embolic events (TEE) in critically ill patients with severe COVID-19 infection. This protocol could therefore be used in a help-for-decision protocol in order to initiate or no anticoagulation.
- When abbreviations are used, spell out the full word at first mention in the text followed by the abbreviation in the parentheses. Thereafter, use the abbreviation throughout. For example, spell out the full word of LDH in abstract not at line 138;
- Line 50-51: check the sentence because it is not grammatically correct
- Please check the text because in some cases the closing brackets are missing, in other cases the number of patients is not written in letters
some sentences are not grammatically correct
Reviewer 2 Report
This is interesting work in the COVID-19 anticoagulation arena, however, there are some shortcomings that I would address, as outlined below:
- What was the rationale for including the endocan in your risk score? What was the biological and pathophysiological ratio for this? This biomarker's underlying biology and imputed significance should be elaborated on in greater detail.
- I would suggest the authors include pertinent references citing the work of Battistoni et al. regarding the impact of chronic and in-hospital outcomes on survival among hospitalized patients with COVID-19. Please see the ref: Battistoni I, Francioni M, Morici N, Rubboli A, Podda GM, Pappalardo A, Abdelrahim MEA, Elgendy MO, Elgendy SO, Khalaf AM, Hamied AAM, Garcés HH, Abdelhamid OES, Tawfik KAM, Zeduri A, Bassi G, Pongetti G, Angelini L, Giovinazzo S, Garcia PM, Serino FS, Polistina GE, Fiorentino G, Barbati G, Toniolo A, Fabbrizioli A, Belenguer-Muncharaz A, Porto I, Ocak S, Minuz P, Bernal F, Hermosilla I, Borovac JA. Pre- and in-hospital anticoagulation therapy in coronavirus disease 2019 patients: a propensity-matched analysis of in-hospital outcomes. J Cardiovasc Med (Hagerstown). 2022 Apr 1;23(4):264-271. doi: 10.2459/JCM.0000000000001284. PMID: 34878430.
- Do authors think that using their risk-stratification and decision-making provided by the results of CALT1 and CALT2 scores would be somewhat limited given that in many systems routine measurement (or any at all) of endocan would be unavailable? This would render the inability to use this score in many practices worldwide. This should be regarded as a limitation.
- What would be the diagnostic value and strength of your scores if endocan would be omitted from the score? This should be elaborated.
- Lack of validation cohort makes these findings limited.
- The anticoagulation regimen should be better defined. When authors say "to initiate anticoagulation" what exactly do they mean by that? Is it prophylactic anticoagulation? Full-dose therapeutic anticoagulation? Which agents are they referring to? This should be strictly defined.
Reviewer 3 Report
This work aims to predict TEE and guide the usage of anticoagulation therapy in severe COVID-19. Although COVID-19 pandemics has past, the data and methods may still be useful for other viral infection. However, a few concerns shall be addressed:
In the table 2: Fibrinogen, D-dimer, LDH and ferritin are all significantly different between the groups, but stepwise multiple variable analysis for CALT1 only retains ferritin. The authors shall discuss this issue to rule out any artificial effects or human errors.
In CALT2, only LDH is not retained and sound more reasonable. It would be better to combine CALT1 and CALT2 together to give an overall analysis to see what happens. If the resultant variables close to CALT2, it might be due to the small case number of TEE within the 0-48h that bias the outcomes.
In future, it would be better to have a verification cohort from different hospitals to demonstrate the developed formulae is practical and useful.
Well written, only minor correction is required
Round 2
Reviewer 2 Report
Thank you for addressing all my comments. No further questions.